# Effect of Plastic Deformation and Acidic Solution on the Corrosion Behavior of Ti-6Al-4V ELI Titanium Alloy

Xuyong Zheng [1], Chen Xu [1], Yi Cai [2],*, and Binbin Zhang [1],*

[1] College of Mechanical and Electrical Engineering, Wenzhou University, Wenzhou 325035, China; 21451439050@stu.wzu.edu.cn (X.Z.); 21461440087@stu.wzu.edu.cn (C.X.)

[2] TeJi Valve Group Co., Ltd., Wenzhou 325102, China

* Correspondence: teji@teji-valve.com (Y.C.); zhangbinbin@wzu.edu.cn (B.Z.)

**Abstract:** This study investigated the tensile deformation of Ti-6Al-4V ELI titanium alloy and its effect on corrosion performance. The results showed that the structural morphology of the samples' strain levels of 0%, 5%, and 10% had minimal changes under an optical microscope. Further investigation of grain orientation information was conducted using electron backscatter diffraction (EBSD), revealing that tensile deformation induced grain rotation, resulting in the diversity of originally preferred orientation grains and a decrease in texture strength. A small amount of {10–12}<−1011> extension twinning formed during the tensile deformation process. The electrochemical properties of Ti-6Al-4V ELI samples with different strain levels were evaluated in 3.5% NaCl solution with pH values of 7 and 1.5. The results indicated that both plastic deformation and acidic environments were detrimental to the passivation film on the titanium alloy surface, leading to reduced corrosion resistance.

**Keywords:** titanium alloys; microstructure; deformation mechanism; corrosion behavior





## 1. Introduction

Titanium alloys are widely used in marine engineering and medical fields because of their excellent mechanical properties and good corrosion resistance [1–3]. Ti-6Al-4V alloy is the most widely used titanium alloy because of its excellent comprehensive properties. Ti-6Al-4V ELI alloy is a titanium alloy obtained by reducing the interstitial element content on the basis of Ti-6Al-4V. Ti-6Al-4V ELI alloy is also used as shell material for deep-sea submersibles due to its high specific strength and excellent corrosion resistance.

For titanium alloys used in marine engineering and medical applications, there are two factors that affect the service performance of materials, stress and corrosion. Currently, there have been many studies on the tensile deformation mechanism of Ti-6Al-4V ELI alloy. Meng et al. [4] have investigated the anisotropic mechanical behavior of Ti-6Al-4V alloy through tensile tests in different processing sampling directions. They found that the stress distribution of the samples stretched along the rolling direction and transverse direction were different, resulting in different dislocation generation within the grains and thus leading to different plastic deformation. Numerous studies have shown that the deformation mechanism of titanium alloys at room temperature is mainly composed of dislocation and twinning. Zhou et al. [5] found that prismatic slip was the dominant slip system initiated in the bimodal microstructure titanium alloy, while the dislocation multiplication of prismatic slip dominates the equiaxed α grains. Ma et al. [6] observed a large number of {10–12} twins in the fatigue crack tip plastic zone in Ti-6Al-4V alloy.

Moreover, Ti-6Al-4V ELI alloy, when used as a load-bearing component in physiological environments or seawater, inevitably encounters corrosive conditions. Titanium alloy forms a dense passive film, which is mainly composed of $TiO_2$ [7,8]. Corrosion resistance of titanium alloy could be influenced by many factors, such as chemical composition, microstructure, and internal stress deformation. Existing studies have demonstrated that alloying elements Pd and Ni can significantly enhance the corrosion resistance of Ti-6Al-4V

in $H_2SO_4$ solution. Additionally, elements Ni and Mo also influence the electrochemical reactions related to corrosion. Furthermore, the stability of protective passive layers clearly depended on the crystallographic substrate orientation. The corrosion resistance of Ti-6Al-3Nb-2Zr-1Mo alloy in 3.5 wt% NaCl and 5 M HCl solutions is improved due to the decrease in the thickness of both β and α phases [9–11]. However, there are few reports on the influence of stress on corrosion behavior. Guo et al. applied two types of compressive deformation, 1% and 50%, to study the effect of plastic deformation on the corrosion resistance of Ti-23Nb-0.7Ta-2Zr. They found that a low level of plastic deformation is harmful to corrosion resistance, while larger deformations often eliminate this harmful effect. Krawiec et al. investigated the corrosion behavior of Ti-6Al-4V and Ti-10Mo-4Zr alloys in a Ringer's solution after different levels of plastic strain. They found that the current density in both the passivation region and the cathodic region increased compared to the current density of the unstrained samples. Additionally, due to the complex composition of the passive films on these two alloys, the current density in the passivation region was lower than that of pure titanium [12–14].

There are many studies on the effect of crystal orientation on the stress corrosion performance of titanium alloys. Chi et al. [15] studied the stress corrosion behavior of Ti-6Al-4V alloy in simulated saltwater solution and found that α grain orientation affected the crack propagation path, thereby affecting the stress corrosion sensitivity of the titanium alloy. Li et al. [16] studied the stress corrosion behavior of steel in artificial seawater and found that cracks tend to initiate and propagate along the <111> orientation according to the grain orientation map. Frank et al. [17] discovered that the number density of crack initiation sites on nanocrystals was heavily dependent on the crystallographic grain surface orientation. Crystals with a low-index (110) orientation on the surface experienced the most significant inhibition in terms of crack initiation density. Conversely, crystals with low-index {001} and {111} orientations experienced the least inhibition.

The corrosive solution is another important factor affecting the corrosion of titanium alloys. Inflammation in the human body can lead to acidification of body fluids, and the dissolution of titanium at the crack tip during stress corrosion can also contribute to acidification of the solution. Therefore, using acidic solutions can simulate the corrosion behavior at crack tips in metals. Souza et al. [18] conducted a study on the electrochemical behavior of titanium alloys Ti-6Al-4V and Ti-13Nb-13Zr in acidic and neutral Ringer's physiological solution. They found that the corrosion current was highest under acidic conditions, while acid conditions had an influence on the passive film. However, there are limited results regarding the effect of plastic deformation on the corrosion resistance of Ti-6Al-4V ELI alloy, especially in acidic environments.

The purpose of this study is to investigate the effect of tensile deformation and acidic environments on the corrosion behavior of Ti-6Al-4V ELI alloy. The microstructural evolution of Ti-6Al-4V ELI alloy during tensile deformation was investigated using an optical microscope, X-ray diffraction (XRD) and EBSD. Then electrochemical measurements were used to analyze the effect of plastic deformation and acidic solution on the corrosion behavior.

## 2. Materials and Methods

The initial materials were Ti-6Al-4V ELI alloy forging bars with a diameter of 200 mm. Then, they were cut into uniaxial tensile specimens with a diameter of 15 mm using wire cutting. Subsequently, the specimens were subjected to uniaxial tensile tests of 5% and 10% strain using an Instron-1343 universal testing machine. After deformation, the specimens were cross-sectioned in a direction along the loading direction for further testing.

XRD analysis was performed using a D/Max 2500 X-ray diffractometer to determine the crystal structure of the alloy. X-ray diffraction utilized Cu Kα radiation from a source. The scanning step size was 0.02° with a scanning speed of 4°/min. The operating voltage of the X-ray diffractometer was 40 kV, and the operating current was 100 mA. In microstructure analysis, the samples were polished and immersed in Kroll reagent (HF: $HNO_3$: $H_2O$ = 1:2:50), followed by immediate rinsing with water. The microstruc-

ture was observed using an OLYMPUS GX71 metallographic microscope (Olympus Corp., Tokyo, Japan). The samples were further processed using electrochemical polishing. The polishing solution used was a mixture of high-chloric acid, butanol, and methanol solution, with a volume ratio of 10:20:70. Subsequently, the samples before tensile testing and those subjected to uniaxial tension until fracture were analyzed for microstructure using the electron backscatter diffraction (EBSD, Hikari XP, AMETEK, San Diego, CA, USA) technique on a scanning electron microscope (JSM-7900F, JEOL Japan Electronics Co., Ltd, Tokyo, Japan). EBSD data acquisition was performed at an accelerating voltage of 20 kV, with a scanning step size set within the range of 0.1 to 0.5 μm. Finally, TSL OIM Analysis 7 software was used for analysis and processing.

In electrochemical measurements, a platinum sheet and a saturated calomel electrode (SCE) were chosen as the auxiliary electrode and reference electrode, respectively, and the Ti-6Al-4V ELI alloy sample was used as the working electrode. This three-electrode system was connected to the CHI 660E electrochemical workstation (ChenHua instruments Co., Ltd., Shanghai, China) in all the electrochemical measurements. The electrolyte solution was a 3.5% NaCl solution, and the pH value was adjusted to 1.5 using lactic acid. The electrochemical properties of Ti-6Al-4V ELI samples with tensile strains of 0%, 5%, and 10% were measured in both neutral and acidic environments. After obtaining a stable open-circuit potential (OCP) value for 7200 s, electrochemical impedance spectroscopy (EIS) measurements were performed in the frequency range of $10^{-2}$ Hz to $10^5$ Hz, with an AC amplitude of 10 mV. The EIS results were analyzed using ZView 3.1 software. Potentiodynamic polarization tests were carried out between −0.5 V (relative to SCE) and 2.5 V. The scan rate was set at 1 mV/s. CView 3.4 software was used to fit polarization curves and corrosion potential ($E_{corr}$), and corrosion current densities ($I_{corr}$) were obtained. To ensure data reproducibility, at least three parallel experiments were conducted in this study.

## 3. Results and Discussion

### 3.1. Microstructural Evolution

Figure 1 shows the metallographic image of Ti-6Al-4V ELI alloy samples with different strain levels. At 0% strain, there were a large number of equiaxed α grains, as well as interstitial β phases in the form of lamellar α structure embedded within them, as shown in Figure 1a. The size of the equiaxed α phase was approximately 10–30 μm, accounting for 50% of the volume, and the width of the lamellar α phase was about 3–4 μm, indicating a typical bimodal microstructure. After tensile tests, no significant change in optical microstructure could be observed, and the average grain sizes were 11.8 μm, 12.2 μm, and 11.2 μm, with minimal variations in grain size.

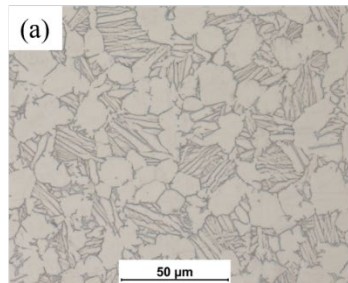 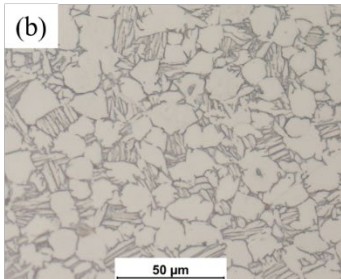 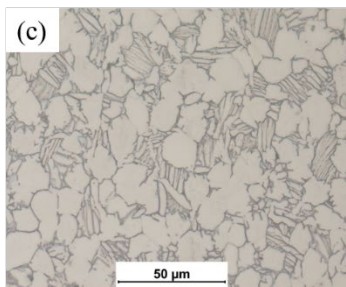

**Figure 1.** Optical micrographs of specimens at different strains: (**a**) 0%; (**b**) 5%; and (**c**) 10%.

The XRD patterns of Ti-6Al-4V ELI alloy samples with different strain levels are shown in Figure 2. All the samples were primarily composed of an α phase and β phase, with a much higher content of the α phase than the β phase. Diffraction peaks shifted to higher 2θ angles, indicating lattice change along the tensile direction. A weak peak broadening was also observed, suggesting the increase in lattice strain and the accumulation of dislocations.

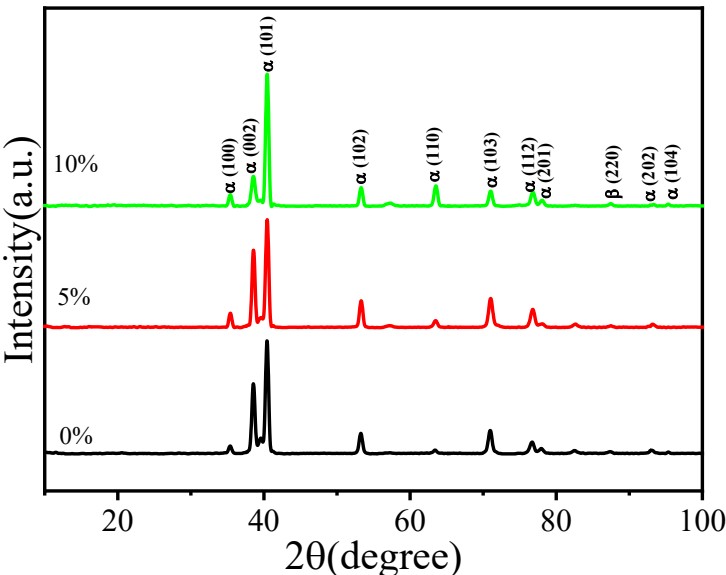

**Figure 2.** XRD patterns of Ti-6Al-4V ELI alloy under different strains.

The microstructure and grain boundary misorientation distribution are shown in Figure 3a, where the blue line represents high-angle grain boundaries (HAGBs, θ > 15°), and the red line represents low-angle grain boundaries (LAGBs, 2° < θ < 15°). Figure 3b shows the phase distribution map in the analyzed area [19–21]. Figure 3c displays the grain boundary misorientation distribution. The proportion of HAGBs was 44%. The LAGBs were concentrated within 5°, and the HAGBs were concentrated between 55° and 65°. Figure 3d presents the statistical analysis of grain size in the α phase of the Ti-6Al-4V ELI alloy, which coincided with the results in the optical micrograph.

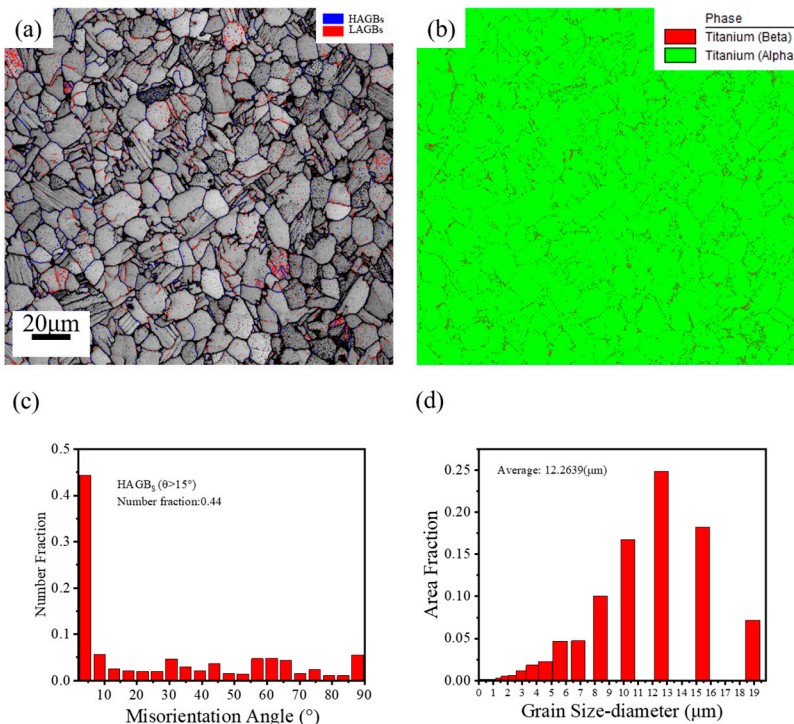

**Figure 3.** Characterization of initial Ti-6Al-4V ELI: (**a**) SEM; (**b**) Phase distribution map; (**c**) Misorientation distribution; (**d**) Grain size distribution.

Figure 4 shows the microstructure of the specimen at different stages of deformation and the corresponding inverse pole figures (IPFs). Figure 4a displays the IPF maps of the sample without any tensile deformation, where the predominant color was red, indicating that most of the grain orientations were close to the <0001> direction. Figure 4b presents the IPF map of the specimen subjected to a tensile strain of 5%. In this case, the predominant color was purple, suggesting that most of the grain orientations were close to the <10–10> direction. Figure 4c shows the IPF map of the specimen subjected to a tensile strain of 10%. At this stage, the grains no longer exhibited preferred orientations, and the distribution of orientations became more scattered [22]. Scattered orientations could be attributed to the loading directions during forging and tensile processes; compressive stress was added in the materials during the forging process, while tensile stress were applied in the tensile test, the opposite direction of stress made the deformation more uniform.

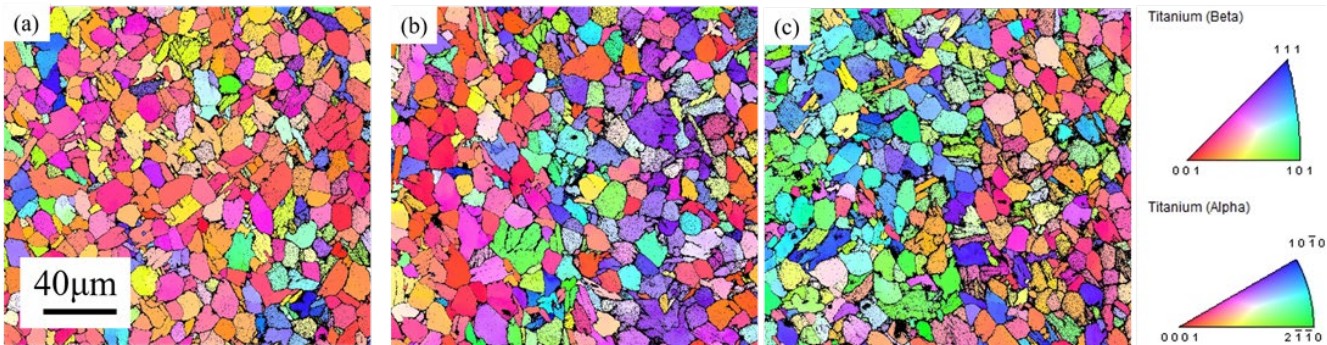

**Figure 4.** IPFs of specimens at different strains: (**a**) 0%; (**b**) 5%; and (**c**) 10%.

Figure 5 shows the pole figures (PFs) of the specimen at different deformation stages. Figure 6a is the pole figure of the α phase in Ti-6Al-4V ELI alloy without strain. Except for the texture with a strength of 17.879 in the {0001} pole figure, which was close to the rolling direction, the textures of other planes were not very pronounced. Figure 5b represents the pole figure of the α phase in Ti-6Al-4V ELI alloy after applying a uniaxial tensile strain of 5%; it could be seen that the texture strength of the α phase was weakened after tensile deformation compared to the sample without deformation, and the maximum strength of the texture component decreased from 17.879 before deformation to 9.249 after deformation. There were some texture components with the highest strength of approximately 9, rotating counterclockwise by about 45° from the A2 axis to the A1 axis around the sample, while the textures of other planes were more dispersed and less pronounced. These grain rotations resulted in changes in the orientation of the grains, leading to variations in the texture components and texture strength of the specimen. Figure 5c depicts the pole figure of the α phase in Ti-6Al-4V ELI alloy after uniaxial tensile deformation with a strain of 10%. From the pole figure, it could be observed that the texture strength of the α phase in Ti-6Al-4V ELI alloy was further weakened compared to the texture strength before deformation, with a maximum reduction of 8.011. Additionally, the texture strength was more dispersed in all three planes, which all coincided with the IPF results. During the tensile deformation process, some α grains rotated to accommodate deformation, causing changes in grain orientation and leading to the diversity of originally preferred orientation grains and finally resulted in a decrease in texture strength.

Figure 6 shows the kernel average misorientation (KAM) maps a of the Ti-6Al-4V ELI alloy samples as a function of the plastic strain. The KAM is often used to represent the local dislocation density. It could be seen that high strain was mainly distributed in lamellar α grains and grain boundaries. And the average KAM values were 1.12°, 1.23°, and 1.40°. The local strains increased with the increase in plastic deformation, which could be attributed to the residual stress caused by the tensile test.

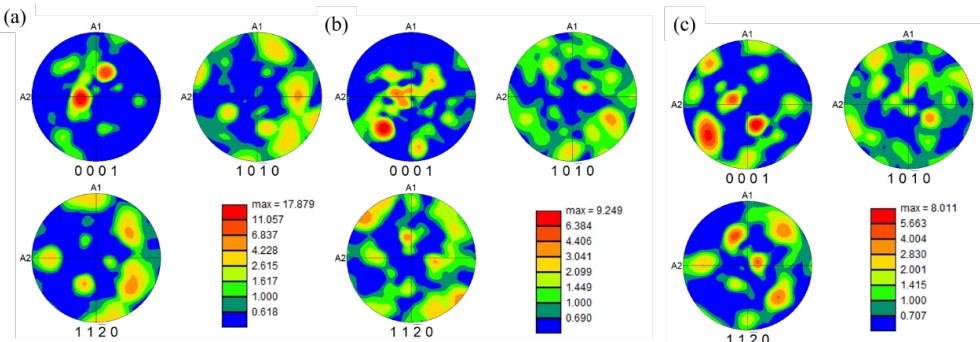

**Figure 5.** PFs of specimens at different strains: (**a**) 0%; (**b**) 5%; and (**c**) 10%.

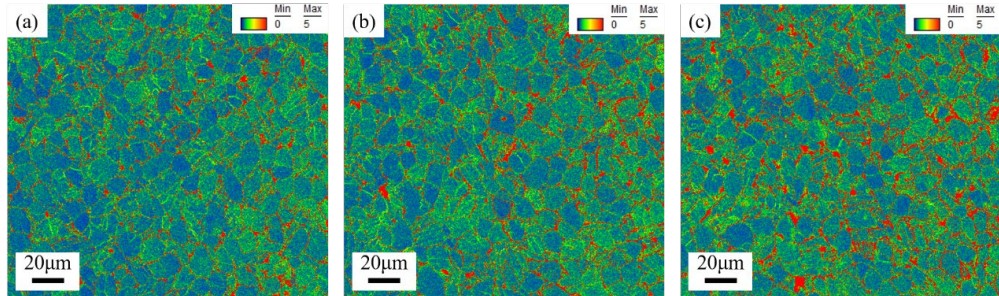

**Figure 6.** KAM maps of specimens at different strains: (**a**) 0%; (**b**) 5%; and (**c**) 10%.

The analysis results of the deformed twins formed during the tensile deformation of Ti-6Al-4V ELI alloy are shown in Figure 7. It could be observed that the crystallographic axes of the deformed twin's unit cell, the a-axis or c-axis, were close to 90° with respect to the c-axis or a-axis of the parent phase unit cell in Figure 7a, respectively. One selected deformed twin was chosen as the research object to study the orientation–misorientation distribution along a straight line at the grain boundary between the parent phase and the twin. The obtained orientation–misorientation distribution is shown in Figure 7b. According to the twin crystallography of titanium, there were {10–12}<−1011> tensile twins with a rotation angle of approximately 85° within the grains and {10–11}<10–1–2> compression twins with a rotation angle of approximately 57°. From the distribution of orientation–misorientation change along the distance, it could be seen that the orientation–misorientation between the parent phase and the formed twin was approximately 87°. Based on this orientation–misorientation, it could be determined that the deformation twin formed during the tensile deformation of the Ti-6Al-4V ELI alloy was a {10–12}<−1011> tensile twin [23–27].

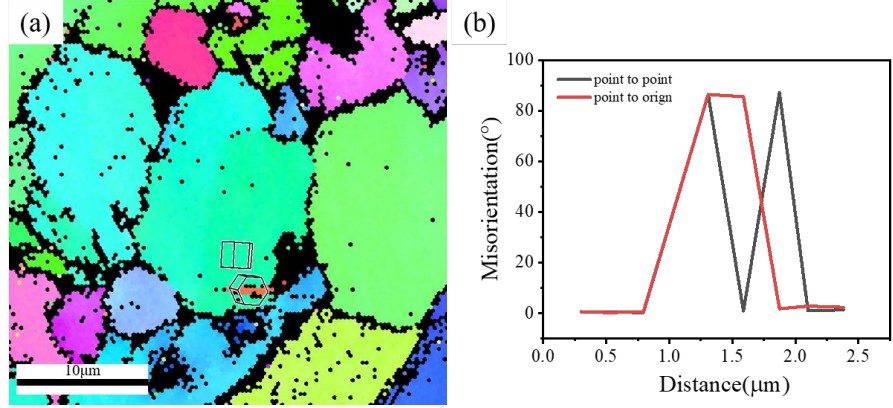

**Figure 7.** Deformation twins in the tensile-strained Ti-6Al-4V ELI alloy: (**a**) Schematic diagram of the twin and parent phase unit cells; (**b**) distribution of orientation–misorientation.

### 3.2. Corrosion Behavior of Deformed Ti-6Al-4V ELI

Figure 8 shows the variation of open-circuit potential (OCP) over time for Ti-6Al-4V ELI alloy immersed in acidic and neutral 3.5% NaCl solutions under different tensile strains. In the 3.5% NaCl solution, as time increased, the OCP values of the specimens gradually shifted towards positive values, indicating the formation of a passive film on the surface of all samples. After a certain period of time, the curves stabilized, indicating the stabilization of the passive film. After immersing for 7200 s, the OCP values of the acidic solution were positive, while the OCP values of the neutral solution were negative, indicating that the quality of the passive films formed on the Ti-6Al-4V ELI alloy was higher in the acidic solution, indicating better corrosion resistance. After 7200 s immersion, the OCP values of 0%, 5%, and 10% samples in acidic solution were about 0.13 V, 0.11 V, and 0.16 V, respectively, as shown in Table 1. Similar trends could be observed in the samples immersed in NaCl solution, and the influence of plastic deformation on the open-circuit potential was not significant.

Figures 9 and 10 represent the Nyquist and Bode plots of the impedance spectra obtained for open-circuit potential for Ti-6Al-4V ELI alloy at different tensile strain levels in different solutions. The Nyquist plots in Figure 9 show relatively large semicircles under all test conditions. Generally, a larger semicircle radius indicated a higher impedance, which meant a more difficult electron transfer between the electrolyte and the tested material [28–31]. Therefore, comparing the conditions, both acidification and deformation decreased the impedance. In the Bode plots, with the increase in deformation, the plateau value of the phase angle in the low-frequency region gradually decreased, and the platform frequency range varies for different samples, indicating non-ideal capacitor behavior. Overall, both deformation and acidification reduced the corrosion resistance of the passive film on the alloy surface. By fitting the EIS results with an equivalent circuit, the capacitance of the passive film formed on Ti-6Al-4V ELI was determined. An electrical equivalent circuit was used to simulate the experimental EIS data, as shown in Figure 9. The parameters of the relevant components in the electrochemical impedance spectrum of the Ti-6Al-4V ELI alloy could be determined by fitting, as shown in Table 2. Here, Rs represented the solution resistance. CPE1 and $R_f$ represented the outer oxide layer capacitance and charge transfer resistance, respectively. CPE2 and $R_{ct}$ represented the double layer capacitance and charge transfer resistance, respectively. The introduction of a constant phase element (CPE) was used to replace the capacitor and describe non-ideal capacitance behavior [32]. The CPE and n represent the constant phase angle element and the degree of deviation of the capacitor, respectively. The EIS results indicated that n was approaching 1, which confirmed the capacitive nature of the oxide films formed on all samples. According to the fitting results, the CPE constants slightly increased with the increasing of strain; the higher capacitance values and lower impedance values point to more active behavior [33]. From Table 2, it could be seen that $R_f$ gradually decreased with increasing strain, the value of $R_f$ decreased from $1.73 \times 10^6$ $\Omega \cdot cm^2$ to $1.80 \times 10^5$ $\Omega \cdot cm^2$. The results indicated that the corrosion resistance of the metal decreased with the increase in deformation and with the decrease in pH, suggesting a thinner passivation film.

**Table 1.** The open-circuit potential (OCP) of the Ti-6Al-4V ELI alloy in the different solutions.

| Sample | pH | OCP (V) |
|---|---|---|
| 0% strain | 7 | $-0.20 \pm 0.11$ |
| | 1.5 | $0.13 \pm 0.08$ |
| 5% strain | 7 | $-0.17 \pm 0.03$ |
| | 1.5 | $0.11 \pm 0.9$ |
| 10% strain | 7 | $-0.18 \pm 0.12$ |
| | 1.5 | $0.16 \pm 0.21$ |

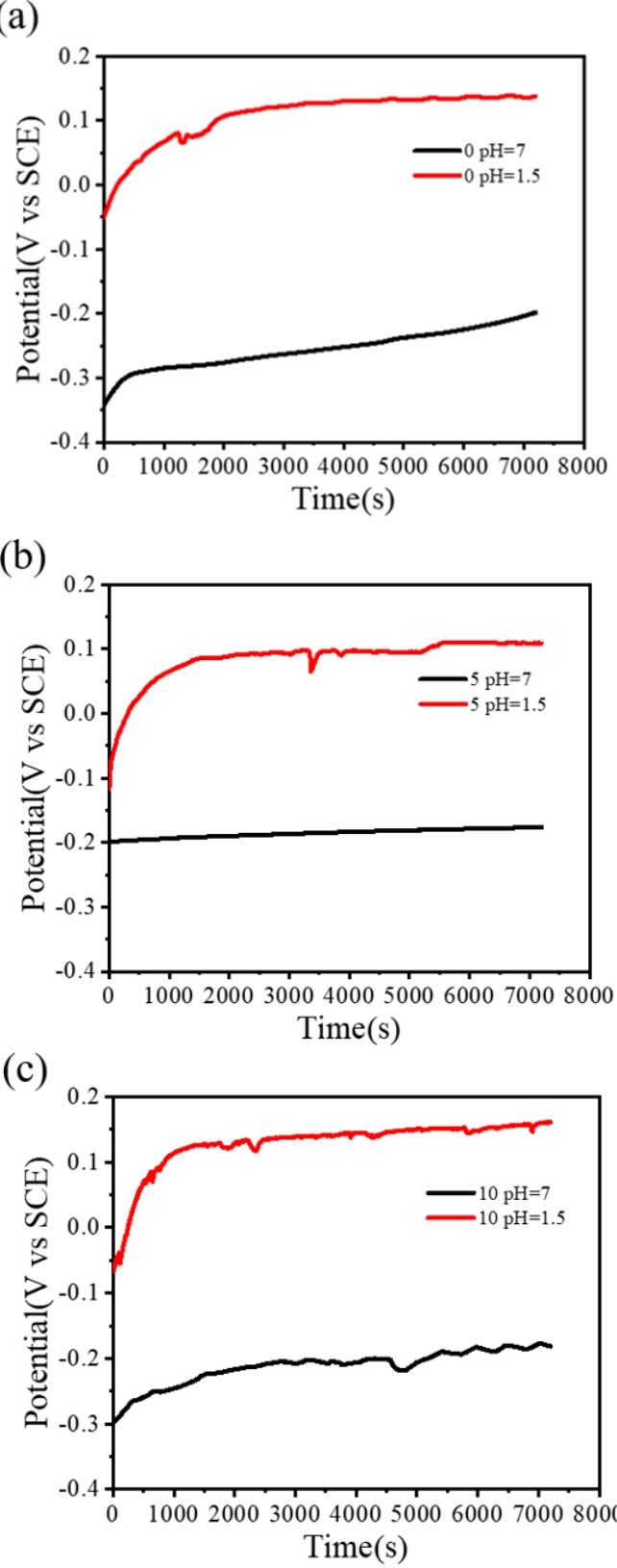

**Figure 8.** The open-circuit potential for Ti-6Al-4V ELI alloy in acidic and neutral 3.5% NaCl solutions under different strains: (**a**) 0%; (**b**) 5%; and (**c**) 10%.

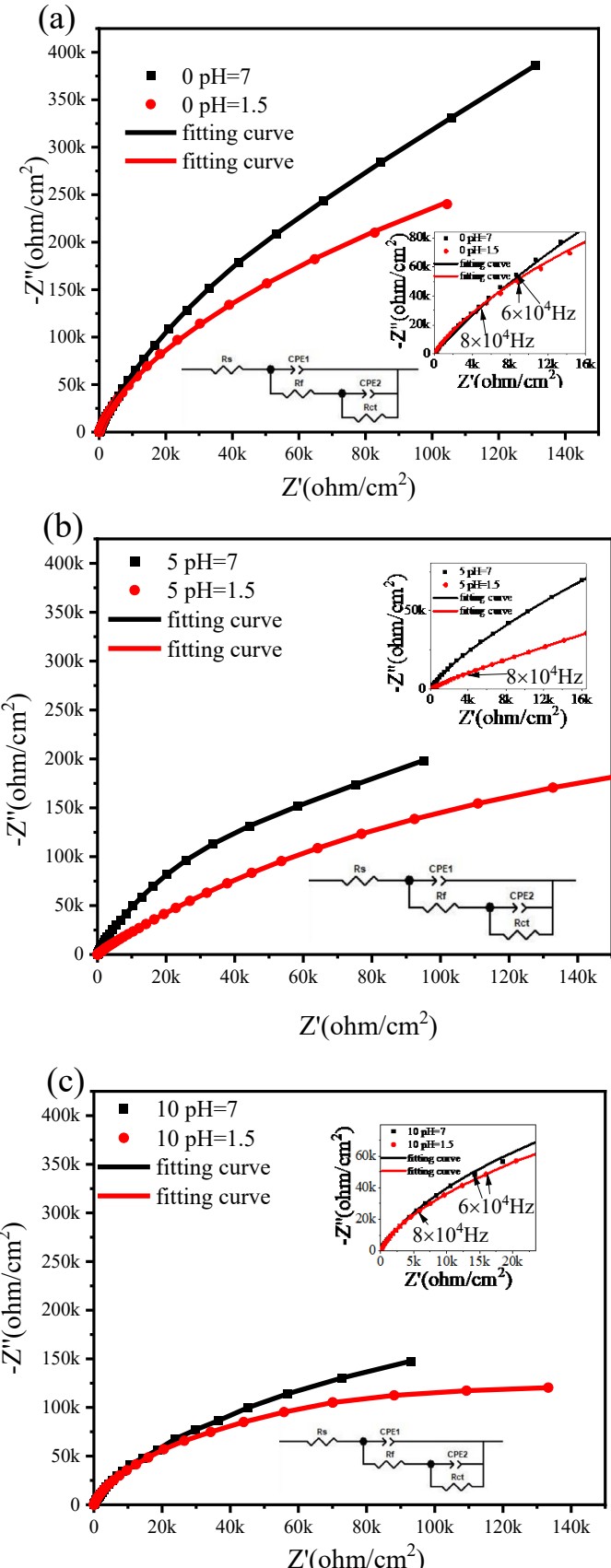

**Figure 9.** The Nyquist plots of Ti-6Al-4V ELI alloy in acidic and neutral 3.5% NaCl solutions under different strains: (**a**) 0%; (**b**) 5%; and (**c**) 10%.

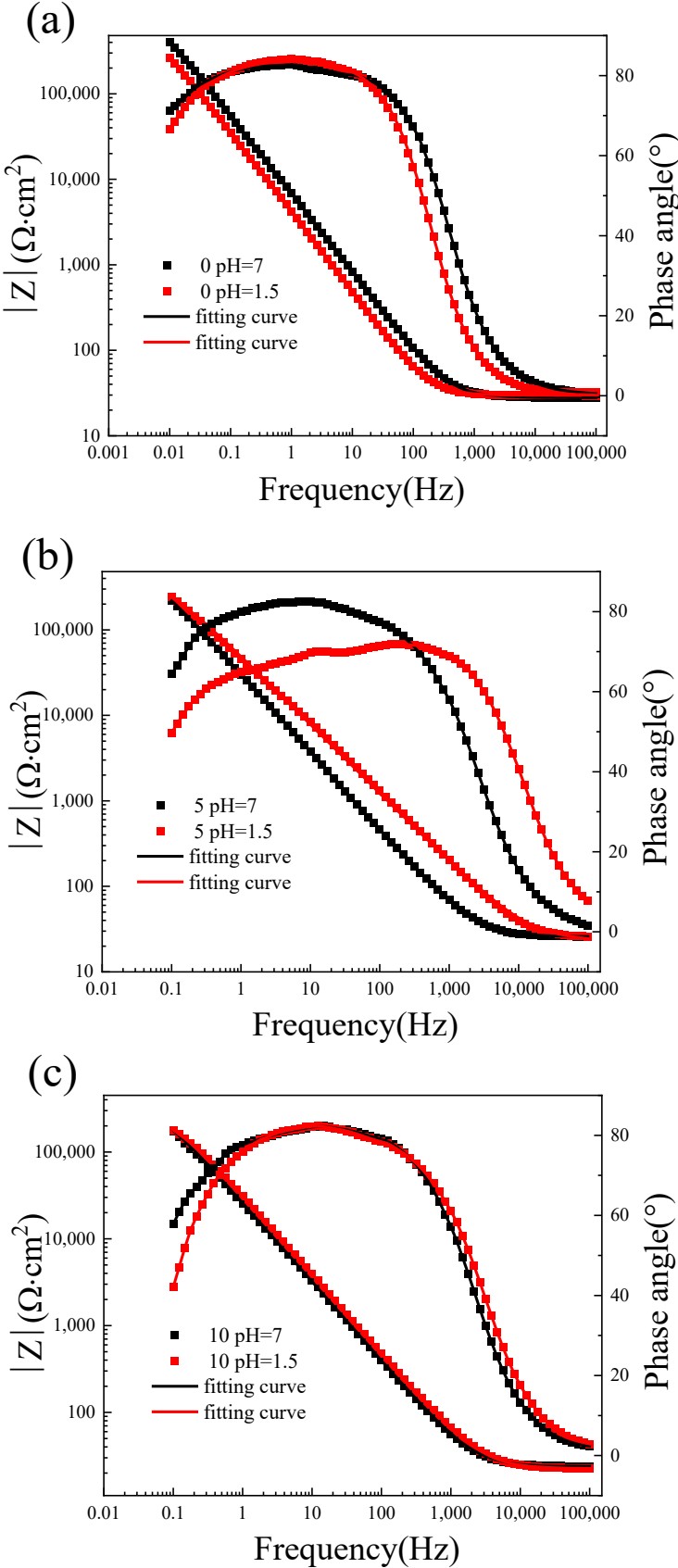

**Figure 10.** The Bode plots of Ti-6Al-4V ELI alloy in acidic and neutral 3.5% NaCl solutions under different strains. Under different strains: (**a**) 0%; (**b**) 5%; and (**c**) 10%.

**Table 2.** The fitting EIS data for the Ti-6Al-4V ELI in the different solutions.

| Sample | pH | $R_s$ ($\Omega \cdot cm^2$) | CPE1 × $10^{-5}$ ($\Omega^{-1} \cdot cm^{-2} \cdot s^n$) | $n_1$ | $R_f$ ($M\Omega \cdot cm^2$) | CPE2 × $10^{-5}$ ($\Omega^{-1} \cdot cm^{-2} \cdot s^n$) | $n_2$ | $R_{ct}$ ($M\Omega \cdot cm^2$) | $\chi^2$ × $10^{-3}$ |
|---|---|---|---|---|---|---|---|---|---|
| 0% | 7 | 28.03 ± 0.53 | 2.74 ± 1.03 | 0.91 ± 0.01 | 1.73 ± 0.07 | 2.24 ± 0.61 | 0.97 ± 0.01 | 4.65 ± 0.61 | 0.23 |
| strain | 1.5 | 29.76 ± 1.02 | 4.26 ± 0.99 | 0.93 ± 0.01 | 0.42 ± 0.08 | 0.73 ± 0.39 | 0.83 ± 0.02 | 1.04 ± 0.55 | 0.25 |
| 5% | 7 | 26.09 ± 0.69 | 5.03 ± 0.24 | 0.90 ± 0.02 | 0.78 ± 0.04 | 0.22 ± 0.07 | 0.87 ± 0.02 | 2.14 ± 0.67 | 1.24 |
| strain | 1.5 | 24.51 ± 0.93 | 2.06 ± 0.54 | 0.85 ± 0.02 | 0.10 ± 0.02 | 1.07 ± 0.02 | 0.85 ± 0.01 | 7.09 ± 0.73 | 0.65 |
| 10% | 7 | 23.89 ± 0.84 | 5.62 ± 0.91 | 0.91 ± 0.01 | 0.18 ± 0.02 | 1.41 ± 0.51 | 0.88 ± 0.01 | 0.29 ± 0.85 | 0.34 |
| strain | 1.5 | 22.67 ± 2.07 | 4.83 ± 0.41 | 0.90 ± 0.02 | 0.28 ± 0.01 | 8.22 ± 0.22 | 0.85 ± 0.01 | 1.14 ± 0.24 | 0.77 |

The polarization curves of Ti-6Al-4V ELI alloy with different strains in 3.5 wt% NaCl solution at different pH values are shown in Figure 11. Under acidic conditions, the strain had a greater influence on the polarization curves compared to the neutral solution. As the strain of the alloy increased in the acidic solution, the corrosion potential ($E_{corr}$) changed from −0.25 V to −0.05 V and finally became +0.09 V when the external strain was 10%. The change from a negative corrosion potential to a positive value was due to a significant increase in cathodic reaction. The corrosion current density ($I_{corr}$) increased fourfold from an initial value of 0.025 µA/cm$^2$ to 0.116 µA/cm$^2$. In the neutral saline solution, this change was not as significant. Due to the large differences in polarization curves under various conditions, some samples could passivate at potential above 0.5 V in acidic solutions, while others failed to form a stable passive film. However, they all passivated at high potential. When the potential was higher than 1.50 V and lower than 2.50 V, the current density immediately entered a region that remained essentially unchanged. At this point, the magnitude of the passive current density was in the order of 10 µA/cm$^2$.

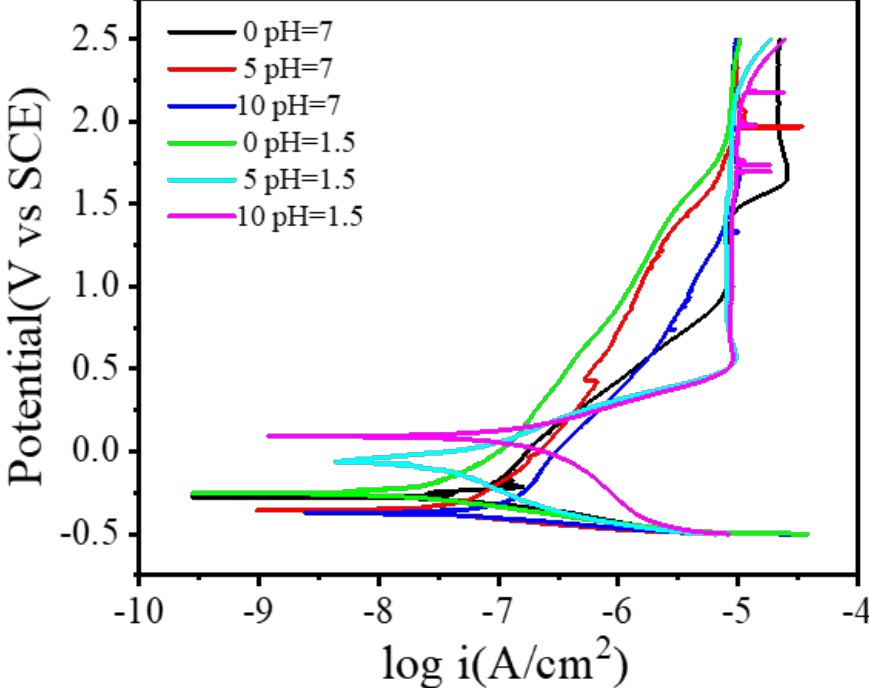

**Figure 11.** Polarization curves of Ti-6Al-4V ELI alloy with different strains in acidic and neutral 3.5% NaCl solutions.

Polarization curves of Ti-6Al-4V ELI under plastic deformation and acidic environments showed significant differences compared to normal conditions. Most studies indicated that titanium alloys initially passivate at lower anodic potential in saline solutions, where the current remains constant as the potential increases [34–36]. During this period, there might be a slight increase in current followed by re-passivation. The oscillation in behavior may be attributed to changes in thin film structure and composition [37]. It was

found that the inability to passivate at low potential was related to plastic deformation within the metal and the acidic environment, aligning with the findings of Krawiec et al. and Munirathinam et al. [38].

Overall, the samples with the highest applied strain showed the worst corrosion resistance. As we know, the passivation film with ceramic structure was incompatible with the deformation of the metal. As the plastic deformation increased, the O content in the passive film became lower, and the passive film became more defective and less protective [39]. In electrochemical experiments, it manifested as larger capacitance and smaller resistance, as shown in Tables 2 and 3. Furthermore, a large amount of slip and twinning occurred after deformation, the emergence of slip bands increased the cathodic reactions [40], and repassivation became much slower after the passive film breakdown because it was inhibited by the dislocations [41,42]. Further, the acidic environment accelerated the cathodic process and promoted the dissolution/formation dynamic equilibrium of the passive film [43], and the diffusion coefficient and donor density of the passivation film increased, resulting in higher activity of the passivation film. Therefore, when coupled with acidic environments and stress, the corrosion resistance of the sample was further compromised. As a result, corrosion inhibitor coatings for Ti-based alloys may be an important research direction in the future. Moreover, it is essential to employ corrosion-resistant coatings with high plasticity to ensure that the coating does not crack under large plastic deformation.

**Table 3.** Electrochemical parameters for the Ti-6Al-4V ELI alloy in the different solutions.

| Sample | pH | $E_{corr}$ (V) | $I_{corr}$ ($\mu A/cm^2$) |
|---|---|---|---|
| 0% strain | 7 | $-0.27 \pm 0.39$ | $0.092 \pm 0.21$ |
| | 1.5 | $-0.25 \pm 0.37$ | $0.025 \pm 0.19$ |
| 5% strain | 7 | $-0.35 \pm 0.46$ | $0.040 \pm 0.28$ |
| | 1.5 | $-0.05 \pm 0.16$ | $0.031 \pm 0.01$ |
| 10% strain | 7 | $-0.37 \pm 0.49$ | $0.058 \pm 0.31$ |
| | 1.5 | $0.09 \pm 0.02$ | $0.116 \pm 0.15$ |

## 4. Conclusions

In order to analyze the tensile deformation mechanism and its impact on the corrosion behavior of Ti-6Al-4V ELI titanium alloy, specimens were subjected to 5% and 10% strains and compared with specimens without strain. Additionally, three types of specimens were immersed in acidic and neutral 3.5% NaCl solutions to measure their corrosion behavior. The conclusions were obtained as follows:

- During the tensile deformation process, some α grains underwent rotation to coordinate the deformation. The rotation of these grains resulted in changes in grain orientation, which in turn caused variations in the texture components, and texture strength decreased. Deformation twins could also form within the grains, with the twinning generated during deformation being {10–12}<−1011> tensile twins;
- The titanium alloy without any tensile strain exhibited the best corrosion resistance, while the titanium alloy subjected to the highest tensile strain showed the poorest corrosion resistance in acidic solutions. Plastic deformation and acidification both led to a decrease in the corrosion resistance of the alloy's passive film on the surface;
- Plastic deformation had a greater effect on the reduction in the impedance of the passive film compared to the acidification of the solution. Due to the interaction of plastic deformations and the acidic environment, the samples failed to form a stable passive film at low anodic potential.

**Author Contributions:** Conceptualization, Methodology, Validation, Writing—original draft, X.Z.; Data analysis, C.X.; Methodology, Writing—review and editing, curation, Formal analysis, B.Z. and Y.C. All authors have read and agreed to the published version of the manuscript.

**Funding:** This work was supported by the Major Science and Technology Special Project in Wenzhou (no. ZF2022002).

**Data Availability Statement:** Data presented in this article are available upon request from the corresponding author.

**Conflicts of Interest:** The authors declare no conflict of interest.

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
