# Peer review of "Effect of Plastic Deformation and Acidic Solution on the Corrosion Behavior of Ti-6Al-4V ELI Titanium Alloy"

_metals, doi:10.3390/met13101740_

Round 1
Reviewer 1 Report
Dear authors and Editors,
I will send a shortly review:
Impedance measurement suffers from poor explanation of phenomena that take place in the system.
Firstly, the authors should provide (enlarge) Nyquist spectra of the high-frequency area.
From the Nyquist plots, it seems that the system has more than a one-time constant (especially for the Figure 9c).
In fact, the value of n (in Table 1.) does not fit the Nyquist spectra. When n=1, the phase angle is -90 degrees.
The authors should comment change in capacity (or Q) for the various strains.
Nyquist plots have on imaginary axes value of the (ohm/cm)^2???
The author must provide Figures obtained from the polarization measurements
Reviewer 2 Report
The corrosion behavior of titanium alloys is an important issue in relation with their uses, especially when they are used for structural applications. To establish dependences between the corrosion resistance and microstructural features of titanium-based materials is a research work worthy to be noted. Even more when the properties of the electrochemical medium are considered, too. The present article contains valuable experimental results regarding the relationship between corrosion resistance and the microstructural features of plastically deformed Ti6Al4V alloy together with pH of electrochemical medium. However, there are some comments, as follows.
2. Materials and methods
a) the authors stated at p.2, lines 82-84 the followings: ” the specimens were cross-sectioned in a direction perpendicular to the loading direction for further testing.”; from this it is understanding that for all corrosion tests, the corroded surface is a perpendicular cross-sectioned one to the loading direction; why the authors did not investigated other surfaces obtained by longitudinal sectioning of samples?
b) the authors stated at p. 3, lines 111-112 that for ”To ensure data reproducibility, at least three parallel experiments were conducted in this study.”; for this reason and to provide a high level of confidence for experimental results, the authors should indicate in manuscript, together with sample’s mean, the standard deviation, too; also, having in view the comparative feature of the results, the authors should include in the manuscript the statistical significance of the null hypothesis testing regarding the equality of mean values for the all quantities experimentally determined.
3. 1. Microstructural evolution
c) in the optic images of the microstructure, the ”lamellar α phase” are ”white”, and ” β phase” (not phases) are dark.
d) the statement from p. 3, lines 125-126 (”Therefore, the applied strain had little influence on the microstructure.”) should be eliminated from the manuscript because it is in contradiction with the statements which follows.
e) in Figure 4, the authors should enlarge the image of the color scale of the orientation map;
3.2. Corrosion Behavior of Deformed Ti-6Al-4V ELI
f) the authors should indicate the OCP values in a table, so that the reader to observe easier and understand the relationship between the deformation and acidity and the corrosion resistance;
g) the authors should include in the Table 1, the values for chi squared test and %error for each element from the equivalent circuit;
h) also, the authors should have in view the comment b.
i) from the manuscript, the effect of the deformation and the effect of the interaction between deformation and acidity is poorly put in evidence; in other words, the authors should revise the manuscript so that they explain why a thinner passivation film was formed with increasing deformation, and with decreasing pH, and the corrosion resistance decreased; what was different in the passivation process?
Reviewer 3 Report
Review Report
Effect of plastic deformation and acidic solution on the corrosion behavior of Ti-6Al-4V ELI titanium alloy
Reading this research shows that it has good results for investigating the effect of plastic deformation and acidic solution on the corrosion behavior of Ti-6Al-4V ELI titanium alloy. The research is written in a smooth and clear English language and the review of the results was very good. The results extracted from this article will help to understand the behavior of Ti-6Al-4V ELI titanium alloy in engineering and medical fields.
· The abstract provided a straightforward summary of all aspects of the study.
· The introduction was skillfully written and covered the main points and significance of the study while citing numerous sources.
· The experimental section was clear and used useful techniques.
· The discussion is meticulously written, explaining all findings, and citing all relevant sources.
· The figures and Tables are beautifully arranged and illustrated and include all the details.
I recommended it to publish.
Reviewer 4 Report
This work has a lot of potential and it is relevant, since not many authors have combined tensile stress and corrosion of alloys. However, it is written in a very poor english, which obscures and makes this work less understandable. The writing style needs to be significantly improved. Below some suggestions to improve the content of the paper:
1) In order The authors should include state of the art and high quality previous works where a detailed study of the crystal orientation enlightens the mechanism of stress corrosion in metallic alloys. For instance, they can mention the work by F.U. Renner et al., Star-shaped crystallographic cracking of localized nanoporous defects, Advanced Materials 27, 4877 (2015).
2) In the experimental section, the brand of the potentiostat should be added, as well as a paragraph in which the authors describe the electrochemical measurements performed (potentiodynamic runs, OCP, etc.)
3) In the results and discussion the authors start by 'This section may be divided by subheadings. It should provide a concise and precise 114 description of the experimental results, their interpretation, as well as the experimental 115 conclusions that can be drawn.'. This paragraph should be erased
4) The EBSD maps are not clear at all, the quality of figure 4 should be improved to be able to read the grain orientations
5) Figure 11 is missing!
6) The authors should explain in detail how they calculate the corrosion current density and voltage and refer to figure 11 (which is actually missing)
7) The authors should discuss on the eventual use of corrosion inhibitor coatings for Ti-based alloys and discuss whether tensile stress experiments would be relevant
English should be significantly improved (writing style and verb conjugation)
Round 2
Reviewer 1 Report
The manuscript is improved according to reviewer comments.
Author Response
谢谢您的夸奖和鼓励。

Reviewer 2 Report
The quality of the revised form of the article is improved one, and the conclusion are much more well sustained by the experimental results.
Author Response
Thank you for your encouragement

Reviewer 4 Report
publish in present form
Author Response
Thank you for your encouragement.
